# Liver Fibrosis during Antiretroviral Treatment in HIV-Infected Individuals. Truth or Tale?

**DOI:** 10.3390/cells10051212

**Published:** 2021-05-15

**Authors:** Athanasios-Dimitrios Bakasis, Theodoros Androutsakos

**Affiliations:** Department of Pathophysiology, Medical School, National and Kapodistrian University of Athens, 11527 Athens, Greece; th.bacasis@gmail.com

**Keywords:** antiretroviral treatment, ART, liver fibrosis, nucleoside reverse transcriptase inhibitors, zidovudine, didanosine, protease inhibitors

## Abstract

After the introduction of antiretroviral treatment (ART) back in 1996, the lifespan of people living with HIV (PLWH) has been substantially increased, while the major causes of morbidity and mortality have switched from opportunistic infections and AIDS-related neoplasms to cardiovascular and liver diseases. HIV itself may lead to liver damage and subsequent liver fibrosis (LF) through multiple pathways. Apart from HIV, viral hepatitis, alcoholic and especially non-alcoholic liver diseases have been implicated in liver involvement among PLWH. Another well known cause of hepatotoxicity is ART, raising clinically significant concerns about LF in long-term treatment. In this review we present the existing data and analyze the association of LF with all ART drug classes. Published data derived from many studies are to some extent controversial and therefore remain inconclusive. Among all the antiretroviral drugs, nucleoside reverse transcriptase inhibitors, especially didanosine and zidovudine, seem to carry the greatest risk for LF, with integrase strand transfer inhibitors and entry inhibitors having minimal risk. Surprisingly, even though protease inhibitors often lead to insulin resistance, they do not seem to be associated with a significant risk of LF. In conclusion, most ART drugs are safe in long-term treatment and seldom lead to severe LF when no liver-related co-morbidities exist.

## 1. Introduction

The human immunodeficiency virus (HIV) infection constitutes a major global public health issue, affecting 37 million people worldwide, of whom 34 million are adults [1]. In the early HIV pandemic, the dominant causes of death were infections such as pneumocystis jirovecci pneumonia (PCP) and acquired immune deficiency syndrome (AIDS) related neoplasms, including lymphomas or Kaposi sarcomas. However, after the introduction of antiretroviral treatment (ART) back in 1996, the leading causes of morbidity and mortality among people living with HIV (PLWH) in industrialized countries have switched to non-AIDS related events, especially cardiovascular and liver disorders [2,3,4,5]. As far as liver involvement is concerned, its prevalence ranges from 4 to 18% according to different studies [6,7,8,9,10,11,12,13,14] while liver-related deaths occur 10 times more often among PLWH compared to the general population. In the D:A:D study involving a large cohort of European, U.S., and Australian HIV-infected patients, 13% of the recorded deaths were liver-related, similar to the 15% reported by the Swiss HIV Cohort Study, although in the latter study this percentage was raised to 18% after the deaths due to hepatocellular carcinomas (HCC) were incorporated [5,8,15,16].

Following the efficacious control of the HIV infection, the causes of liver involvement in PLWH have changed from opportunistic infections and lymphomas to hepatitis C (HCV) and hepatitis B (HBV) co-infections, drug-induced liver injury (DILI), as well as alcoholic (AFLD) and non-alcoholic fatty liver disease (NAFLD) [15,16,17,18]. NAFLD especially affects a growing number of PLWH [17,18,19,20,21] since eating habits, drugs used in ART and HIV infection per se promote liver steatosis, steatohepatitis and subsequent liver fibrosis (LF), cirrhosis and hepatocellular carcinoma [19,20,21,22,23,24].

Several studies have explored the possible causes of liver fibrosis in HIV-infected patients, with most of them pointing out viral hepatitis co-infections, age, and alcohol abuse as the major risk factors [16,25,26,27,28,29,30,31,32,33,34,35,36,37,38,39,40,41]. On the other hand, the long-term hepatotoxicity of ART has been debated, with various studies remaining inconclusive. In this review, the potential pathophysiological pathways leading to liver damage and the existing data regarding the real risk of LF for each antiretroviral drug will be discussed.

## 2. Liver Fibrosis in HIV-Infected Individuals

Apart from those already mentioned, other factors associated with significant LF appear to be chronically elevated serum alanine aminotransferase (ALT) and gamma glutamyl-aminotransferase (γGT) levels, while findings concerning body-mass index and the presence of hypertriglyceridemia or diabetes mellitus (common predisposing factors for NAFLD) are inconsistent [26,28,30,32,33,41,42,43,44,45,46,47]. 

The most interesting finding in all the recent studies is the association between HIV itself and LF. Even though HIV is primarily found in liver macrophages, it may also infect hepatocytes, leading to low-level intrahepatic replication and possibly hepatocyte cell death and inflammation [48,49,50]. In this context, HIV can increase the intra-hepatic expression of procollagen alpha-1 via the activation of the HIV co-receptors C-X-C chemokine receptor type 4 (CXCR4) and C-C chemokine receptor type 5 (CCR5) found on the hepatocyte surface [51,52]. Another potential mechanism of LF is the enhanced expression of the transforming growth factor b1 (TGF-b1) by hepatocytes as a direct effect of the HIV infection [53]. On the other hand, hepatic stellate cells (HSC) can be also be infected, promoting collagen I expression, secretion of the pro-inflammatory chemokine monocyte chemoattractant protein-1 (MCP-1), and the activation of tissue-inhibitor metalloproteinases (TIMPs) [54,55,56]. Last but not least, HIV causes endoplasmic reticulum (ER) stress, leading to mitochondrial toxicity and increased oxidative stress, both resulting in liver injury, decreased beta-oxidation of fatty acids and the accumulation of fat in the liver [57].

HIV may also lead to LF through high amounts of circulating lipopolysaccharides (LPS) [58,59]. More specifically, HIV causes a gut barrier breach and selective CD4 + T-cell depletion enteropathy, resulting in an increased LPS release that may reach the liver though portal circulation. In turn, LPS activates TLR4 on Kupffer cells and induces the overproduction of tumor necrosis factor-a (TNF-a), promoting inflammation and fibrogenesis. Apart from its direct fibrogenic effect, circulating LPS augments insulin resistance (IR) and liver triglyceride accumulation, leading to non-alcoholic steatohepatitis and LF [60].

Similarly, several studies have shown that HIV seropositivity per se and a low CD4 + cell count among PLWH have been associated with a higher degree of LF, irrespective of the presence or absence of viral hepatitis co-infection [61,62,63,64,65,66,67].

## 3. ART and Liver Fibrosis

Assessing the contribution of various ART drugs to LF among PLWH has proven a difficult task through the years, even though the risk of hepatotoxicity and the pathophysiological pathways associated with LF for each ART drug class are largely known (Table 1). As already mentioned, HIV leads to LF via multiple pathways; the effective control of HIV infection seems to be beneficial for liver histopathology. Proof of this comes from the START trial which involved more than 4500 patients, and showed that LF is rather rare in HIV infected individuals with early ART initiation, implying a protective role for early therapeutic intervention on LF progression [30]. In another study by Ding et al. including 3900 patients, ART was correlated with an improvement of LF as evaluated by the fibrosis-4 (FIB-4) score [31], while Thorpe et al. found that ART interruption in 541 HIV/HCV co-infected patients led to a higher probability of liver fibrosis [68]. Similarly, Blackard et al. in a large study of 1227 women, including 312 mono-infected and 498 HIV/HCV co-infected patients, concluded that LF was positively correlated with HIV RNA levels and was negatively correlated with both a CD4 + cell count and ART [65]. Similarly, Mariné-Barjoan et al. described a positive correlation between the time interval from HIV seropositivity to ART initiation, and LF [62]. Moreover, Tural et al. found in a cohort of 126 HIV/HCV co-infected patients that the duration of ART was negatively correlated with LF, and a low CD4 + cell count was positively correlated with LF [69]. Lastly, Chou et al. showed in a study of 228 HIV/HCV co-infected individuals that ART initiation improved LF in patients with a significant LF level at the baseline [70].

## 4. Nucleoside Reverse Transcriptase Inhibitors (NRTIs)

NRTIs were the first anti-HIV drug class; zidovudine (AZT) was the first FDA approved medicine for HIV treatment back in 1987 [71]. After AZT, eight more NRTIs have been approved for treating the HIV infection, including didanosine (DDI) and tenofovir alafenamide (TAF), which gained FDA approval last, in 2015. NRTIs act by inhibiting HIV replication through blocking HIV RNA reverse transcriptase and have been proven to be extremely efficacious against HIV, constituting an integral part of every ART regimen. However, NRTIs, especially those of the ‘first generation’, such as AZT, DDI, zalcitabine (ddC) and stavudine (d4T), exhibit a variety of adverse events, from dyslipidemia and insulin resistance to peripheral neuropathies, myopathies, lactic acidosis and severe hepatotoxicity [71,72]. Additionally, long term exposure to DDI has been associated with non-cirrhotic portal hypertension, a rare but serious complication [73,74,75,76,77].

Mitochondrial toxicity and hypersensitivity reactions are the main mechanisms of NRTIs-induced liver damage. Mitochondrial toxicity is caused by the inhibition of mitochondrial DNA (mtDNA) replication due to the binding of NRTIs to the mitochondrial DNA polymerase gamma. This leads to an impairment of oxidative phosphorylation and promotes the formation of reactive oxygen species which in turn damage mtDNA, resulting in mitochondrial dysfunction [78]. NRTIs also adversely affect the oxidation of free fatty acids within hepatic mitochondria, which, in combination with NRTIs-induced insulin resistance and dyslipidemia, lead to triglycerides accumulation in the liver and subsequent hepatic steatosis [79,80,81,82].

NRTIs are the best studied drugs as far as LF is concerned (Table 2). In two large studies of HIV/HCV co-infected patients, the use of NRTIs was correlated with a worsening of LF, a higher risk of liver decompensation and increased mortality, while in another similar study from McGovern et al., the use of NRTIs, and especially DDI and d4T, were linked to liver steatosis and fibrosis [83,84,85].

However, not all NRTIs carry the same risk for LF; DDI and AZT have the greatest risk. In a large study of 1785 PLWH with a 2-year follow up, higher FIB-4 values were correlated with DDI use, older age, male gender, a low CD4 + cell count, and an unsuppressed HIV viral load [86]. In another study, previous use of DDI was associated with worse LF as estimated by FIB-4 [66]. Merchante et al., after analyzing data from 258 PLWH with no HCV or HBV co-infections assessing LF with TE, supported the claim that the duration of DDI use along with age, increased alcohol intake, previous abacavir (ABC) exposure, and a CD4 + cell count <200 cells/mL were independently associated with significant LF [87]. Another study by Loko et al., comprising of 671 HIV/HCV co-infected patients, showed that using DDI for more than 5 months, together with male gender, a high homeostatic model assessment (HOMA) value, an intravenously acquired HCV infection and lipodystrophy were predisposing factors for significant fibrosis in TE [88]. Similarly, Bani-Sadr et al. showed that DDI use led to a worsening of LF assessed by a liver biopsy in a cohort of 383 HIV/HCV co-infected patients under pegylated interferon and ribavirin treatment [89]. Apart from DDI, AZT has been also associated with LF. In an observational study involving 333 PLWH, LF was evaluated with TE, FIB-4 and the AST to Platelet Ratio Index (APRI) and it was found that DDI and AZT use, HCV co-infection and ongoing HIV replication were significantly correlated with LF [25].

Boyd A et al. also showed a progression of LF under AZT treatment in 167 HIV/HBV co-infected patients [90]. Interestingly, in the same study tenofovir disoproxil (TDF) use did not lead to an improvement of liver fibrosis despite the efficient control of HBV infection. In a large study by Ding et al. with 3500 PLWH, of whom 2675 were HIV mono-infected, TDF was a negative predictor for LF improvement, as assessed by FIB-4 [31]. On the contrary, Vinikoor et al., supported the claim that TDF use was associated with an improvement of LF as measured by TE, irrespective of HBV-co-infection [91]. Likewise, other studies with HIV/HBV co-infected patients found TDF to be beneficial for liver fibrosis [92,93,94,95]. Overall TDF seems to be protective in LF, especially in patients with HIV/HBV co-infection. However, the level of LF with the FIB-4 score may be overestimated in TDF treated patients, since the FIB-4 scoring system incorporates the aspartate aminotransferase (AST) value which may rise in clinical or subclinical muscle damage due to TDF use.

Although many studies have focused on DDI, AZT and TDF, there is insufficient data for the rest of the NRTIs regarding their potential to induce LF. In a study of 112 HIV/HCV co-infected patients undergoing a liver biopsy, it was shown that the use of d4T led to liver steatosis and that patients with steatosis were more prone to develop worse LF. A combination of ABC and lamivudine (3TC) as backbone therapies were shown to increase the APRI score in a cohort of 314 HCV/HIV co-infected Canadian patients while exposure to ABC alone was found to have a negative impact on liver stiffness in a cohort of PLWH without co-infections [87,96,97]. Finally, Emtricitabine (FTC) has not been linked to hepatotoxicity and is considered safe in terms of LF [98].

## 5. Non-Nucleoside Reverse Transcriptase Inhibitors (NNRTIs)

NNRTIs are non-competitive inhibitors of reverse transcriptase affecting the catalytic function of the enzyme by binding to specific tyrosine residues located near the active site [99]. The most common mechanisms attributed to NNRTI hepatotoxicity include a hypersensitivity reaction occurring early during the treatment course and an idiosyncratic late-onset toxicity of metabolic origin; both seem to be more common in patients with HBV or HCV co-infections, although this finding is not constant in all studies [100,101,102,103,104,105]. Not all NNRTIs carry the same risk for hepatotoxicity and patients receiving nevirapine (NVR) and efavirenz (EFV) display a risk of up to 18% and 8%, respectively [106].

Regarding the association of NNRTIs with LF, data are scarce (Table 2). In a cross-sectional study of 152 HIV/HCV co-infected patients, NVR use was found to induce severe LF, although the duration of the exposure was not associated with LF severity [107]. On the contrary, in a study of 201 HIV/HCV co-infected patients whose LF was assessed by a liver biopsy, NNRTIs and especially NVR were found to be associated with a low probability of significant LF, a finding not observed in patients treated with PIs [108]. Likewise, in another study by Fernández-Montero et al. including 545 HIV/HCV co-infected patients, NVR was found to be protective against LF progression [109]. The safety of EFV in patients with liver cirrhosis was studied in a cohort of 189 HIV/HCV co-infected patients, of whom 56 had severe LF and 25 had liver cirrhosis. EFV was proven to be quite safe, with few cases presenting severe transaminitis, low rates of EFV discontinuations due to liver-related events and no deaths due to liver disease progression [110]. To further complicate the role of NNRTI in LF, Rilpivirine (RPV) use in mice ameliorated LF through a selective STAT 1-dependent induction of apoptosis in hepatic stellate cells, which exerted paracrine effects in hepatocytes promoting liver regeneration [111].

## 6. Protease Inhibitors (PIs)

PIs are peptidomimetic molecules that target the active site of HIV aspartic protease, the enzyme responsible for cleaving the precursor viral polyprotein, gagpol, into its constituent proteins. The inhibition of viral polyprotein cleavage results in the production of immature non-infectious viral particles. All FDA-approved PIs have been associated with cases of hepatitis even though the exact mechanism is largely unknown [104]. However, this drug-induced hepatitis usually resolves itself spontaneously and progressive liver damage occurs rarely [112,113,114,115,116].

First generation PIs—e.g., indinavir and ritonavir—are known to induce IR, mainly through dramatically increasing the central adiposity and altering the plasma lipid profile via lipogenesis inhibition, decreasing the hepatic clearance of very-low-density lipoproteins (VLDL), and increasing the production of hepatic triglyceride [117,118,119]. Although the new-generation PIs seem to have little impact on lipid levels as monotherapy, they contribute to an unfavorable lipid profile compared to most other classes of ART when combined with ritonavir or cobicistat in the context of a pharmacological booster [120,121,122,123].

No cases of severe PI-related LF have been reported in the literature, even though IR seems to be a common adverse effect in clinical practice and cases of indinavir-induced hepatitis have been described [124,125,126]. In a study by Fernandez-Montero et al., the use of PIs, mainly Lopinavir, was associated with the progression of LF in a cohort of 545 HIV/HCV co-infected patients, while in a study by Sagir et al., the duration of PI use was positively correlated with LF [109,127]. On the contrary, in a study by Benhamou et al., comprising of 182 HCV/HIV co-infected patients, PLWH using PIs were found to have less LF [128]. Likewise, in a study by Macias et al., the use of PI-based ART led to less LF when compared to no ART [129]. Finally, ritonavir was not found to cause severe hepatotoxicity in a cohort of 117 HCV/HIV co-infected patients, 71 of whom had significant LF, while in another study, switching from ritonavir-boosted PI-based ART to raltegravir-based ART led to a lower level of liver steatosis but not LF [130,131]. The most important studies regarding PIs and LF are summarized in Table 2.

## 7. Integrase Strand Transfer Inhibitors (INSTIs)

INSTIs target the integration process of the HIV viral DNA into the host DNA, a process achieved through a series of DNA cutting and joining reactions, mediated by the retroviral enzyme integrase [132,133]. Current INSTIs target the strand transfer step of the integration process by binding to the active enzyme site and disengaging it from the viral DNA [134]. Five INSTIs are currently in use against HIV with the most recent one, Cabotegravir, obtaining FDA approval in January 2021 as a monthly injection [135,136]. Even though all INSTIs are largely metabolized in the liver by glucuronidation following urinary clearance, they have little or no effect on microsomal cytochrome P450 enzymes and their mechanism of hepatotoxicity is unknown [137].

Overall, INSTIs are considered safe and potent ART drugs; severe transaminitis is a rare event [138,139]. Weight gain under INSTIs treatment has been noted, especially in PLWH treated with dolutegravir (DTG), however the impact on liver steatosis remains controversial, since a study from China correlated the use of INSTIs with steatosis, but two other studies showed an improvement in liver steatosis after switching from PIs to ART including INSTIs [130,140,141,142,143].

Given the lack of data in the literature relating the use of INSTIs to significant LF, and that hepatotoxicity is a rare adverse event of INSTIs, it appears that INSTIs are a safe therapeutic option for patients with liver diseases.

## 8. Entry Inhibitors

Entry inhibitors act by preventing HIV from entering into the host cell [144]. So far, the following entry inhibitors have been approved for HIV treatment and are currently used in clinical practice: (a) Maraviroc (MVC), a C-C chemokine receptor-5 (CCR-5) inhibitor which prevents the interaction of CCR-5 with envelope glycoprotein GP-120 (gp120); (b) Enfuvirtide, a ‘fusion inhibitor’ which binds to the transmembrane glycoprotein GP-41 (gp41) preventing the outer membrane of HIV from fusing to the approximate membrane of T-cells and the subsequent cell entry; (c) Idalizumab which binds to domain 2 of CD4 + T-cells and interferes with the post-attachment steps required for the entry of HIV particles into the host cells; and (d) Fostemsavir, approved by the FDA in 2020, which binds directly to gp120 prohibiting the interaction needed between the virus and the surface receptors on CD4 + T-cells [145,146].

All entry inhibitors have proven to be liver-friendly, even in cirrhotic patients; transaminitis or the deterioration of the liver function are rather rare adverse events [147,148,149]. When hepatotoxicity occurs, this is probably due to drug-drug interactions since entry-inhibitors are extensively metabolized in the liver via the CYP 450 system and are a substrate for P-glycoprotein.

Data regarding entry inhibitors and liver fibrosis are few and mainly concern MVC. In an experimental study by Coppola et al., the addition of MVC in the hepatic stellate cell line blocked the accumulation of fibrillar collagens and the production of extracellular matrix proteins along with a down-regulation of metalloproteinases 2 and 9 (MMP-2, MMP-9) and their inhibitors (TIMP-1, TIMP-2) [150]. In another study by Rossetti et al. in a cohort of 150 patients, switching from an MVC-free 3 drug regimen ART to MVC plus Darunavir/ritonavir led to a better APRI score after 48 weeks (Table 2) [151].

**Table 2 cells-10-01212-t002:** Clinical studies connecting a specific antiretroviral drug or class with liver fibrosis.

First Author, Year	Type of Study	Nr of pts	Assessment of LF	ART Drug/Class Correlated with LF	Outcome
McGovern BH, 2006 [85]	Retrospective	183	LB	NRTIs	NRTIs, especially DDI and d4T, associated with liver steatosis and fibrosis
Sulkowski MS, 2005 [97]	Cross-sectional	112	LB	NRTIs	d4T associated with liver steatosis and eventually LF
Focà E, 2016 [84]	Retrospective	1433	FIB-4	NRTIs NNRTIs PIs	Prolonged exposure to NRTIs predicted LF progression; possible protective effect of NNRTIs and PIs
Merchante N, 2010 [87]	Prospective	258	TE	DDI	Duration of DDI use associated with significant LF
Loko MA, 2011 [88]	Prospective	671	TE	DDI	Use of DDI for more than 5 months predisposed for significant LF
Kapogiannis BG, 2016 [86]	Prospective	1785	FIB-4 APRI	DDI	DDI use correlated with a worse FIB-4 and APRI scores
Kooij KW, 2016 [66]	Cross-sectional	598	FIB-4	DDI	Prior use of DDI associated with worse LF measured by FIB-4 score
Anadol E, 2018 [25]	Cross-sectional	333	APRI FIB-4 TE	DDI	History of exposure to DDI associated with significant LF and cirrhosis
Boyd A, 2017 [90]	Prospective	167	Fibrotest	AZT	LF progression under AZT treatment
Brunet L, 2016 [96]	Prospective	314	APRI	Abacavir, 3TC	Abacavir/3TC as backbone associated with higher APRI
Vinikoor M, 2017 [91]	Prospective	463	TE	TDF	TDF use associated with improvement in TE score
Ding Y, 2017 [31]	Retrospective	3900	FIB-4	TDF	TDF was a negative predictor for LF improvement
Benhamou Y, 2001 [128]	Retrospective	182	LB	PIs	PIs associated with lower LF stage
Macías J, 2004 [107]	Cross Sectional	152	LB	PIs NVR EFV	NVR use associated with severe LF; duration of exposure not correlated with LF grade PI-based ART led to less LF when compared to no ART
Fernández-Montero JV, 2014 [109]	Retrospective	545	TE	PIs (mainly Lopinavir) NVR	NVR associated with protection and PI use (mainly Lopinavir) associated with progression of LF
Berenguer J, 2008 [108]	Cross Sectional	201	LB	NNRTIs PIs	NNRTIs (especially NVR) associated with low probability of significant LF
Macías J, 2017 [140]	Prospective	39	TE	EFV, Raltegravir	Switching Efavirenz to Raltegravir showed decreases in the degree of hepatic steatosis
Calza L, 2019 [130]	Prospective	61	TE FIB-4	PIs	Change from ritonavir-boosted PI-based ART to raltegravir-based ART led to a lower liver steatosis but not LF grade
Rossetti B, 2019 [151]	Prospective	150	FIB-4 APRI	MVC	Switch from MVC-free to MVC+Darunavir/ritonavir led to a better APRI score after 48 weeks

**Abbreviations:** 3TC: Lamivudine; APRI: AST to platelets ration index; AZT: Zidovudine; DDI: Didanosine; d4T: Stavudine; EFV: Efavirenz; FIB-4: Fibrosis-4 score; LB: Liver Biopsy; LF: Liver fibrosis; MVC: Maraviroc; NNRTIs: Non-Nucleoside reverse transcriptase inhibitors; NRTIs: Nucleoside reverse transcriptase inhibitors; NVR: Nevirapine; Protease Inhibitors; TDF: Tenofovir Disoproxil Fumarate; TE: Transient elastography.

## 9. Conclusions

ART has changed the landscape of HIV infection, increasing the lifespan of PLWH and changing the major causes of morbidity and mortality. However, concern has risen regarding the short and long-term hepatotoxicity of ART. Several studies have attempted to address this issue, but many controversies still exist, since the same drugs have been found to carry a significant risk for LF in one study, and minimal or no risk in another.

According to most studies, ‘first generation’ NRTIs, mainly DDI and AZT, may lead to significant LF after long-term treatment, a possibility that is decreasing with ‘new generation’ NRTIs, such as TDF or FTC. On the other hand, NNRTIs have not been implicated into fibrogenesis, with the exception of NVR, while the same applies for INSTIs. Surprisingly, even though PIs are known to cause insulin resistance and weight gain, data are controversial with some studies showing an improvement while others showing a worsening of LF under PI-containing ART regimens. Lastly, entry inhibitors seem to carry a minimal risk for hepatotoxicity or significant LF.

In conclusion, a reasonable therapeutic strategy is to avoid using ‘first generation’ NRTIs in PLWH with predisposing factors for LF or established, significant LF, and choose ‘new generation’ NRTIs or other ART-class drugs instead.

## Figures and Tables

**Table 1 cells-10-01212-t001:** Antiretroviral drug classes and the proposed mechanisms of liver damage associated with their use.

ART Drug Classes	Mechanism of Liver Damage
NRTIs	Mitochondrial toxicity Liver steatosis Hypersensitivity reactions
NNRTIs	Mitochondrial toxicity Hypersensitivity reactions
PIs	Liver steatosis Direct drug toxicity (rare) Hypersensitivity reactions
INSTIs	Hepatotoxic metabolic by-products
Entry inhibitors	Drug-drug interactions Hypersensitivity reactions

Abbreviations: ART: Antiretroviral treatment; NRTIs: Nucleoside reverse transcriptase inhibitors, NNRTIs: Non-nucleoside reverse transcriptase inhibitors, PIs: Protease inhibitors, INSTIs: Integrase strand transfer inhibitors.

## Data Availability

Not applicable.

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
