# Peer review of "Liver Fibrosis during Antiretroviral Treatment in HIV-Infected Individuals. Truth or Tale?"

_cells, 2021, doi:10.3390/cells10051212_

Round 1
Reviewer 1 Report
Overall the authors are to be commended for their paper discussing this important topic. They cover quite a bit of the relevant literature in a useful and organised way.
There are a few instances in which the referencing is not clear (see below), and some phrasing is a bit misleading.
Specifically -
- Introduction - paragraph 1 '34 of them' (presumably 34 million?);
- Introduction - para 3. 'studies failing to establish the accurate risk' (should be referenced or clarified)
- In section (2) para 2 'HIV does not replicate in hepatocytes' - this needs to be referenced (ref 36, 37 show expression of ligands but no evidence for/against hepatocyte replication)
- In section (2) para 3 - 'HIV can' lead to liver fibrosis (perhaps 'may' - causality of this not proven? and need references here to show).
- section 2 - final para - 'led to' is misleading I think - prob 'associated with'
- Section 4 - para 1 'a few years ago' suggest to be more specific - state year
- Section 4 para 4 - 'accused for LF' - suggest rephrase
- Section 4 para 5 - 'mainly due to suppression of HBV' (how do we know that? do we know that?)
- Table 2 Ref Anadol et al - 'cirrhosis' spelling mistake
- Conclusion - final sentence is unclear and poorly worded
Author Response
Thank you very much for your kind words. Following you will find our responses to your comments. Note that changes can be found in the manuscript highlighted in yellow.
- Thank you for your comment. 34 million is the correct phrase and this has now been added.
- Thank you for your comment. We have rephrased the 'studies failing to establish the accurate risk' to ‘the long-term hepatotoxicity of ART has been a long debate, with various studies showing various results’. We have not referenced this sentence since it is thoroughly discussed in the following sections of the article.
- Thank you very much for your remark. You are right and in fact HIV seems to show low-level intrahepatic replication. We have now rephrased the sentence 'HIV does not replicate in hepatocytes' to ‘the virus seems to also infect hepatocytes, leading to low-level intrahepatic replication and possibly hepatocyte cell death and inflammation’ and have also referenced it.
- Thank you for your comment. ‘May’ has replaced ‘can’ in the sentence and references have been added.
- Thank you for your comment. We have replaced 'led to' with 'associated with'
- Thank you for your remark. We have replaced 'a few years ago' with the year TAF was approved from the FDA for use in HIV-infected individuals.
- Thank you for your suggestion. We have rephrased 'accused for LF' with ‘associated with’
- Thank you very much for your remark. Even though in a study by Stockdale A et al (Liver fibrosis by transient elastography and virologic outcomes after introduction of tenofovir in lamivudine-experienced adults with HIV and hepatitis B virus (HBV) coinfection in Ghana) the decline in HBV DNA in a subset of HIV/HBV infected patients under TDF treatment led to an improvement of transient elastography irrespectively of HIV detection rates, it is clear that no sufficient data exist to support this phrase and therefore it is now deleted.
- Thank you very much. Cirrhosis is now spelled correctly.
- Thank you very much for your remark. We agree that the last sentence was unclear. We have therefore deleted it and have added ‘and choose ‘new generation’ NRTIs or other ART-class drugs instead’ in the final sentence of our document. We have also rechecked the quality of written english and have added a few, new studies in our manuscript. You can find these new studies on table 2 and within the text (section 4, paragraphs 4 and 5, section 6, paragraph 3). As a result of these new studies we have changed the phrase concerning PIs in conclusion section.
Reviewer 2 Report
The review by Androutsakos and Basakis is well written and adresses an interesting topic. I have a few suggestions to further improve this interesting read:
- Many of the references are more than 10 years old.
- References around the impact of alcohol use on liver firbrosis in HIV-infected patients are lacking.
- The authors should reference the study by Blackard and colleagues that used FIB-4 to assess liver fibrosis in a cohort of HIV-infected women (https://pubmed.ncbi.nlm.nih.gov/21248367/).
- The authors should mention that NRTIs were also associated with the presence of non-cirrhotic portal hypertension in some series.
Author Response
Thank you very much for your comments. Below you will find the answers to your comments. Note that changes can be found in the manuscript highlighted in yellow.
- Thank you very much for your comment. It is true that some of the studies included in the article are old, so we have added new, more recent, references. Moreover we have added a few, new studies in our manuscript. You can find these new studies on table 2 and within the text (section 4, paragraphs 4 and 5, section 6, paragraph 3). As a result of these new studies we have changed the phrase concerning PIs in conclusion section. We hope you will find the new reference list up-to-date.
- Thank you very much for your remark. Recent studies addressing the impact of alcohol in HIV-infected individuals’ liver fibrosis are now added both in the final paragraph of the introduction as well as in the first paragraph of section 2.
- Thank you very much for your suggestion. We have added this study as a reference in the last paragraph of section 2. We have also discussed this study in section 3, paragraph 1 as it is a large study, highlighting the importance of HIV infection control in preventing liver fibrosis.
- Thank you for your comment. We have added NCPH as an adverse effect of prolonged DDI exposure in section 4, paragraph 1 and have added appropriate references.